# CROWDSOURCED PHRASE-BASED TOKENIZATION FOR LOW-RESOURCED NEURAL MACHINE TRANSLATION: THE CASE OF FON LANGUAGE

## ABSTRACT

Building effective neural machine translation (NMT) models for very low-resourced and morphologically rich African indigenous languages is an open challenge. Besides the issue of finding available resources for them, a lot of work is put into preprocessing and tokenization. Recent studies have shown that standard tokenization methods do not always adequately deal with the grammatical, diacritical, and tonal properties of some African languages. That, coupled with the extremely low availability of training samples, hinders the production of reliable NMT models. In this paper, using Fon language as a case study, we revisit standard tokenization methods and introduce Word-Expressions-Based (WEB) tokenization, a human-involved super-words tokenization strategy to create a better representative vocabulary for training. Furthermore, we compare our tokenization strategy to others on the Fon-French and French-Fon translation tasks.

## 1    INTRODUCTION

We would like to start by sharing with you this Fon sentence: « mɛtà mɛtà wɛ zìnwó hɛn wa aligbo mɛ ».

How would you tokenize this? What happens if we implement the standard method of splitting the sentence into its word elements (either using the space delimiter or using subword units)?

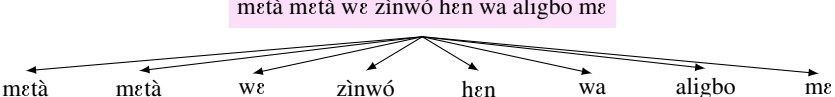

Well we did that and discovered that a translation (to French) model, trained on sentences split this way, gave a literal translation of «chaque singe est entré dans la vie avec sa tête, son destin (English: each monkey entered the stage of life with its head, its destiny)» for the above Fon sentence. But we are not talking about a monkey here ☺. It is a metaphor and so some of the words should be taken collectively as phrases. Using a phrase-based tokenizer, we got the following:

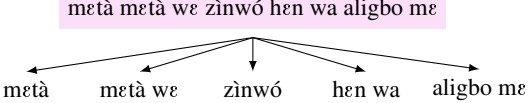

A native speaker looking at some of these grouped phrases will quickly point out that some of the grouped phrases are wrong. Probably the phrase-based model could not effectively learn the phrases due to the low data it was trained on? Also, we got a translation of «singe chaque vient au monde dans vie avec tête et destin (English: monkey each comes into world in life with head and fate)» . However, this translation is still not correct. The expression actually means «Every human being is born with chances» . Another interpretation would be that we must be open to changes, and constantly be learning to take advantages of each situation in life ☺.

One illustrative example, which we encourage the reader to try, is to go to Google Translate and try translating **«it costs an arm and a leg»** to your language (native language or a language you understand). For the 20 languages we tried, all the translation results were wrong: literal and not fully conveying the true (some would say phrasal) expression or meaning. The expression **«it costs an arm and a leg»**, just means **«it is expensive»**. Now imagine a language with a sentence structure largely made up of such expressions – that is Fon ☺.

Tokenization is generally viewed as a solved problem. Yet, in practice, we often encounter difficulties in using standard tokenizers for NMT tasks, as shown above with Fon. This may be because of special tokenization needs for particular domains (like medicine (He & Kayaalp, 2006; Cruz Díaz & Maña López, 2015)), or languages. Fon, one of the five classes of the Gbe language clusters (Aja, Ewe, Fon, Gen, and Phla-Phera according to (Capo, 2010)), is spoken by approximately 1.7 million people located in southwestern Nigeria, Benin, Togo, and southeastern Ghana. There exists approximately 53 different dialects of Fon spoken throughout Benin. Fon has complex grammar and syntax, is very tonal and diacritics are highly influential (Dossou & Emezue, 2020). Despite being spoken by 1.7 million speakers, Joshi et al. (2020) have categorized Fon as «left behind» or «understudied» in NLP. This poses a challenge when using standard tokenization methods.

Given that most Fon sentences (and by extension most African languages) are like the sentence example given above (or the combination of such expressions), there is a need to re-visit tokenization of such languages. In this paper, using Fon in our experiment, we examine standard tokenization methods, and introduce the **Word-Expressions-Based (WEB) tokenization**. Furthermore, we test our tokenization strategy on the Fon-French and French-Fon translation tasks. Our main contributions are the dataset, our analysis and the proposal of WEB for extremely low-resourced African languages (ALRLs). The dataset, models and codes will be open-sourced on our Github page.

## 2 BACKGROUND AND MOTIVATION

Modern NMT models usually require large amount of parallel data in order to effectively learn the representations of morphologically rich source and target languages. While proposed solutions, such as transfer-learning from a high-resource language (HRL) to the low-resource language (LRL) (Gu et al., 2018; Renduchintala et al., 2018; Karakanta et al., 2018), and using monolingual data (Sennrich et al., 2016a; Zhang & Zong, 2016; Burlot & Yvon, 2018; Hoang et al., 2018), have proved effective, they are still not able to produce better translation results for most ALRLs. Standard tokenization methods, like Subword Units (SU) (Sennrich et al., 2015), inspired by the byte-pair-encoding (BPE) (Gage, 1994), have greatly improved current NMT systems. However, studies have shown that BPE does not always boost performance of NMT systems for analytical languages(Abbott & Martinus, 2018). Ngo et al. (2019) show that when morphological differences exist between source and target languages, SU does not significantly improve results. Therefore, there is a great need to revisit NMT with a focus on low-resourced, morphologically complex languages like Fon. This may involve taking a look at how to adapt standard NMT strategies to these languages.

## 3   TOKENIZATION STRATEGIES AND THEIR CHALLENGES FOR FON

In this section, we briefly discuss the standard tokenization strategies employed in NMT, as well as challenges faced while applying them to Fon.

**Word-Based tokenization (WB)** consists of splitting sentences into words, according to a delimiter. We'll show the limits of this method using this Fon expression: *«un ɖo ganji»* . *«un»* on its own is an interjection, to express an emotion of surprise or astonishment. But *«un ɖo»* already means *"I am"*, *"I am at"*, or *"I have"*, depending on the context in which it is used. The whole expression, *«un ɖo ganji»* , could mean "I am fine" or "I am okay".

**Phrase-Based tokenization (PhB)** encodes phrases (group of words) as atomic units, instead of words. As a result, models trained on PhB have the ability to learn and interpret language-specific phrases (noun, verbal and prepositional phrases), making it better than WB for Fon language. However, due to the low-resourcedness of the language and the randomness of PhB alignments, some extracted pairs are not always contextually faultless. For example, the computer alignment gave respectively [zɛn, une (a, an, one)] and [azɔn, la (the)] , instead of [zɛn, une marmite (a pot)] and [azɔn, la maladie (the disease)] .

**Encoding with SU** has made great headway in NMT, especially due to its ability to effectively encode rare out-of-vocabulary words (Sennrich et al., 2016b). Machácek et al. (2018), in analyzing the word segmentation for NMT, reported that the common property of BPE and SU relies on the distribution of character sequences, but disregards any morphological properties of the languages in question. Apart from rule-based tokenization, there are machine learning approaches to tokenization as well, which unfortunately require a substantial amount of training samples (both original and tokenized versions of the same texts) (Riley, 1989; Mikheev, 2000; Jurish & Würzner, 2013). To the best of our knowledge, there is no known language-specific tokenization proposed for ALRLs, although there have been a number of works on adapting NMT specifically to them (like (Orife et al., 2020; van Biljon et al., 2020; Vaswani et al., 2017), to mention but a few).

## 4   WORD-EXPRESSIONS-BASED TOKENIZATION (WEB)

WEB involves aligning and extracting meaningful expressions based on linguistic components of Fon (phonemes, morphemes, lexemes, syntax, and context). This requires the assistance of Fon-French native speakers. Some examples of good alignments are

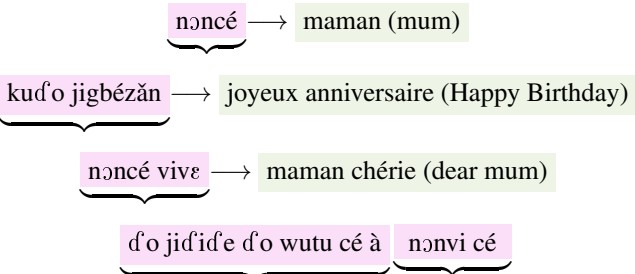

It is important to note that WEB is not a human-in-the-loop process, because it doesn't require human intervention to run. The human intervention occurs while cleaning and preprocessing the dataset. Although not perfect yet, we describe our algorithm as a recursive search algorithm, which looks and finds the most

optimal combination of words and expressions which will produce a better translation for a source sentence. The following algorithm was designed to encode and decode sentences using the established vocabularies:

1. **Run** through the vocabulary and output a list L of all possible word combinations of words and expressions appearing in the sentence $S$.

2. Important principle in Fon: higher word orders = more precise and meaningful expressions. Using this principle, for each element (word or expression), $w \in L$,

   (a) **Check** if there exists a higher word order, $v \in L$, such that $w \subsetneq v$.

   (b) **If** 2a is true, discard w, **else** keep w.

3. The output is a list $\hat{L}$ of optimal expressions that from the initial L, making up the initial sentence $S$.

4. Add <start> and <end> taggers respectively at the beginning and the end of every element $\hat{w}$ (word or expression) $\in \hat{L}$.

5. Encode every $\hat{w}$ (word or expression) $\in \hat{L}$

We argue that WEB scales well because it does not require any linguistic annotations but knowledge and intuitions from bilinguals, meaning, we can crowdsource those phrases. We want to state clearly, in order to avoid any confusion, that WEB is another version of PhB, involving human evaluation. For our study, it took a group of 8 people, all bilinguals speaking Fon and French, and 30 days to align and extract meaningful sentences manually. No preliminary trainings have been done with the annotators, given the fact that they are in majority linguists and natives of the Fon language. This made the step of sentences splitting into expressions, more natural, reliable and faster.

## 5   THE FON-FRENCH DATASET: DATA COLLECTION, CLEANING AND EXPANSION PROCESSES

As our goal is to create a reliable translation system to be used by the modern Fon-speaking community, we set out to gather more data on daily conversations domain for this study. Thanks to many collaborations with Fon-French bilinguals, journalists and linguists, we gathered daily citations, proverbs and sentences with their French translations. After the collection's stage, we obtained a dataset of 8074 pairs of Fon-French sentences.

The cleaning process, which involved the Fon-French bilinguals, mainly consisted of analyzing the contextual meanings of the Fon sentences, and checking the quality of the French translations. In many cases, where the French translations were really bad, we made significant corrections.

Another major observation was the presence of many long and complex sentences. That's where the idea of expanding the dataset came from: we proceeded to split, when possible, Fon sentences into short, independent, and meaningful expressions (expression of 1-6 words), and accordingly add their respective French translations. At the end of these processes, we obtained our final dataset of 25,383 pairs of Fon-French sentences. The experiments, described in this paper, were conducted using the final dataset.

We strongly believe that involving the Fon-French bilinguals into the cleaning process, greatly improved the quality of the dataset. In fact, many initial translation errors were disregarded by standard, rule-based tokenization (like WB, PhB and SU) and cleaning techniques[1]. However, with the help of the **intuitive or natural** language knowledge of the Fon-French bilinguals, bunch of those errors have been fixed. This highlights, the importance of having native speakers of ALRLs to clean and review the dataset, during the initial stages of its compilation.

---

[1]Using Python Regex and String packages (https://docs.python.org/3/library/re.html) together with NLTK preprocessing library (https://www.nltk.org/)

## 6 Methodology, Results and Conclusion

In this section, we describe the implementation of WB, PhB, SU, WEB and we compare the results of our NMT model trained on them for our analysis.

### 6.1 Creation of vocabularies for WB, PhB, SU and WEB

For WB, we split the sentences according to the standard 'space' delimiter, using the TensorFlow-Keras text tokenizer[2], getting a vocabulary of 7,845 and 8,756 Fon and French tokens (words) respectively.

For PhB, we use the IBM1 model from nltk.translate.api module[3] to align and extract all possible pairs of sentences. Our main observation was that, some pairs generated were either not meaningful or not maching, but we didn't try to rearrange them in order to see how well the generated pairs, without human intervention, would affect the translation quality. In so doing, we got a vocabulary of 10,576 and 11,724 Fon and French tokens respectively (word and expressions).

To implement SU , we used TensorFlow's SubwordTextEncoder[4] and built a vocabulary of 7,952 and 8,116 Fon and French tokens (words and subwords) respectively.

To implement WEB, we considered unique expressions as atomic units. Using the steps highlighted for WEB in section 4, we encoded those atomic units and obtained a vocabulary of 18,759 and 19,785 Fon and French tokens (word and expressions) used for the model training.

### 6.2 Dataset splitting, model's architecture and training.

From the dataset, we carefully selected 155 long and complex sentences i.e. sentences made of 5 or more expressions, as test data; sentences that we believe, would test the model's ability to correctly translate higher word order expressions in Fon. 10% of the training data, was set aside for validation.

For training, we used an encoder-decoder-based architecture (Sutskever et al., 2014), made up of 128-dimensional gated rectified units (GRUs) recurrent layers(Cho et al., 2014), with a word embedding layer of dimension 256 and a 10-dimensional attention model (Bahdanau et al., 2015).

We trained with a batch size of 100, learning rate of 0.001 and 500 epochs, using validation loss to track model performance. The training took all the 500 epochs, with the loss reducing from one epoch to another. We would like to emphasize that up only at 500 epochs, with the given hyperparameters, we obtained significant and meaningful translations.

All training processes took 14 days on a 16GB Tesla K80 GPU. We evaluated our NMT models performances using BLEU (Papineni et al., 2002), METEOR (Banerjee & Lavie, 2005), CharacTER (TER) (Wang et al., 2016), and GLEU (Wu et al., 2016) metrics.

### 6.3 Results and Conclusion:

Table 1 and Table 2 show that our baseline model performs better with PhB, and best with WEB, in terms of metric and translation quality. It is important to note that while BLEU scores of PhB and WEB, reduced on the Fr→Fon task, BLEU scores of WB and SU improved on it. We speculate that this might be because WB and SU enhanced the model's understanding of French expressions over Fon, confirming the findings of (Abbott & Martinus, 2018). Ngo et al. (2019). This corroborates our argument that in order to help NMT

---

[2]https://www.tensorflow.org

[3]https://www.nltk.org/api/nltk.translate.html

[4]http://www.tensorflow.org

| Translation | Tokenization | BLEU ↑ | METEOR ↑ | TER ↓ | GLEU↑ |
|---|---|---|---|---|---|
| Fon → Fr | WB | 6.8 | 12.2 | 86.2 | 9 |
| Fon → Fr | SU | 7.6 | 13.6 | 87.4 | 10 |
| Fon → Fr | PhB | 38.9 | 53.7 | 43.9 | 42 |
| Fon → Fr | **WEB** | **66.6** | **77.77** | **24.2** | **67** |
| Fr → Fon | WB | 15.65 | - | - | 8 |
| Fr → Fon | SU | 25.68 | - | - | 9 |
| Fr → Fon | PhB | 38.74 | - | - | 25.74 |
| Fr → Fon | **WEB** | **49.37** | - | - | **43** |

Table 1: Experimental results of our model trained on WB, SU, PhB and WEB.

systems to translate ALRLs better, it is paramount to create adequate tokenization processes that can better represent and encode their structure and morphology.

This is a pilot project and there is headroom to be explored with improving WEB. We are also working on combining WEB with SU, to get the best of both worlds. To promote research and reproducibility in this direction, the dataset and model will be made publicly available on Github after the review. Simultaneously, we are working on releasing platforms for the translation service to be used. We believe that it would be a good way to gather more data and keep constantly improving the model's performance.

| | Sentences: **Fon** , **French and English Translations** |
|---|---|
| Source | a ɖo jiɖiɖe ɖo wutu cé à nɔnvi cé |
| Tokenization output | a ɖo jiɖiɖe ɖo wutu cé à    nɔnvi cé |
| Target | est-ce que tu me fais confiance mon frère? (my brother, do you trust in me?) |
| WB | confiance mon oncle (trust my uncle) |
| PhB | tu me fais confiance? (do you trust me?) |
| SU | aies la foi (have faith) |
| **WEB** | mon frère, est-ce que tu me fais confiance? (my brother do you trust in me?) |
| Source | ɖé é man yɔn nùmi à, na bɔ yi doto hwé |
| Tokenization output | ɖé é man yɔn nùmi à ,    na bɔ yi doto hwé |
| Target | j'irai à l'hopitâl vu que je ne me sens pas bien(Since I am not feeling well, I will go to hospital) |
| WB | être malade et se rendre à l'hopitâl (to be sick and to go to hospital) |
| PhB | je me rends à l'hopitâl parce que je ne me sens pas bien ]coloreng(I am going to hospital because I am not feeling well) |
| SU | rends à l'hopitâl, je suis malade (Go to hospital, I am sick) |
| **WEB** | je me rendrai à l'hopital vu que je ne me sens pas bien (I will go to hospital since I am not feeling well) |

Table 2: Model translations with WB, PhB, SU and WEB

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
