# OpenReview forum: "Crowd-sourced Phrase-Based Tokenization for Low-Resourced Neural Machine Translation: The case of Fon Language"
_ICLR.cc/2021/Conference — Reject_

### Official Review · AnonReviewer2 · 2020-10-27
**Good work, but polish it a bit.**

**Rating:** 5
**Confidence:** 5

**Review:**

Edit after seeing others reviews -- I think I gave this paper a MUCH higher score than the other reviewers, simply because it is very novel with Fon language. I agree with all of your points about what is lacking, but in my mind, the novelty was enough to still give a 7. Now I definitely think that is too high. I think this paper can reasonably be rejected, but I'd like to give actionable of constructive criticism, since I do think the work on this low resource language is important for the NLP community. With such low resources, we cannot expect the same type of work as we would for other languages.

Overview: This paper discusses the problems of common tokenization strategies for low resource african languages, and proposes a new tokenization method to overcome these problems. They train low resource NMTs using 4 different tokenization strategies, to show that their proposed tokenization method leads to the best NMT results by several metrics.

Contribution: The authors contribute a new tokenization method, code, and a dataset.

The good: Very interesting and important work! Many people will be excited to use this data. Paper is mostly clearly written, and easy to read. The paper flows well. Someone with this paper could reproduce the work, more or less.

The bad:
* Figure 1 is difficult to read and messy. First, by "Input" you actually mean "Source". The input would be the source sentence with its appropriate tokenization, no? Also, I think putting the english translation in a different font or color would be greatly helpful to our eyes. I really think this must be fixed! Figure 1 is presently not pleasant to look at, even though it has interesting results`!
* Section 4 - I think you really need to re-state that the algorithm has a human-in-the-loop for clarity. Before describing your algorithm, humans are only mentioned once in the algorithm. Indeed, at first, the words "The following algorithm" confused me, because I thought it was more a "methodology", since Step 2 is where the humans are in the loop, unless you have a Fon POS tagger and I am misunderstanding? But then at the end, I saw you include Encode as step 4, so it is the machine...  The fact that I flustered a bit with my understanding here, was confused, and had to spend a few minutes thinking about it, means it needs a bit of tweaking. Maybe add a comment saying Step 2 is the human-in-the-loop step of the algorithm?

Suggested additions:
* I think more specific linguistic details about Fon are missing. For example, if you could give us one or two sentences of Fon in the beginning of the paper, that demonstrate some of the difficulties of the language, I think this would greatly strengthen the motivation. You *tell* us that Fon is "a language with special tokenization needs" and that "standard tokenization methods do not alwaysadequately deal with the grammatical, diacritical, and tonal properties of some African language", and you cite the relevant papers. But I would still like to be *shown*. I think just including two sentences that have some of these features, and that gets the point accross of "how would we tokenize this?" would really help the motivation. Its not that I/readers dont believe you when we are *told*, but being *shown* makes it much more interesting and give people an appreciation for Fon tokenization challenges!
* Can we get any information about how the annotators were trained? I think this is standard for such papers.

Other smaller suggested fixes:
* Section 5, near the end - Little grammatical mistake. "... bunch of those errors has" should be "errors have".
* Section 6.3 - Please change "The results from Table 2 and Table 1" to say "Table 1 and Table 2". It does not make sense to list them out of order. I also think it makes sense to switch Figure 1 and Figure 2 entirely. I.e., Figure 1 should be your results table, and figure 2 should be the examples for us to see.
* Section 6.3 - Slightly confusing wording. The second sentence is confusing to me, and I am a native English speaker. "It is important to note that while BLEU of other methods reduced on the Fr→Fon task, WB improved on it."  To me saying "BLUE reduced for the other methods" means that you have some other baseline you are comparing to. Am I missing something? Are you comparing against Fon--> Fr?

Questions:
* Section 6.2 - Does it really take all 500 epochs to run, or do you have early stopping at some point when the loss flatlines?
* Because BPE is such a standard baseline, why do you not include it as a baseline? I know you cite the Abbott & Martinus, 2018 paper, stating that BPE is bad for analytical languages, but I still think it would prove a point to show BPE performing badly for your data.

Overall: Very interesting work, and can't wait to see this data be used :-) I think the paper could be greatly strengthened by taking some time to include an example that demonstrates the linguistic and typological features of Fon that makes it difficult.

---

### Official Review · AnonReviewer3 · 2020-10-28
**A "Fon" Project: How to Translate an African Low-resource Language**

**Rating:** 3
**Confidence:** 3

**Review:**

The authors investigate different tokenization methods for the translation between French and Fon (an African low-resource language). This means that they compare different ways to construct the input and output vocabularies of a neural machine translation (NMT) system. They further propose their own way to create those units, based on phrases, which is called WEB.

The NMT system the authors use follows Bahdanau et al. (2015): it is a GRU sequence-to-sequence model with attention. The dataset they use has been created and cleaned by bilingual speakers and consists of roughly 25k examples (this is a really small dataset for NMT, so the authors are taking on a really hard task!).

WEB works in the following way: after phrases have been found automatically, bilingual speakers analyze what the longest phrases which correspond to translated phrases in the other language are. Only the longest phrases for each example are kept for the final vocabulary. The authors show that WEB improves the performance in both translation directions by a lot on all metrics, clearly showing that the work they invest into creating the vocabulary pays out. Thus, I think this work is important to be able to provide speakers of Fon with a functioning translation system.

However, I am unsure if this work is suitable for a machine learning conference. While the overall goal of this work is to create an NMT system, the main contribution is the manual cleaning of the dataset and semi-manual creation of the vocabularies. I would recommend to the authors to submit this paper to a conference with a stronger focus on NLP and NLP resources (maybe LREC)? I further want to emphasize that I think work like this paper is incredibly important and the authors shouldn't feel discouraged. Importantly, the manual labor needed for WEB has been a lot and it's obvious that it helps for NMT. I just don't think that this paper is a good fit for ICLR.

Minor point: has the creation of WEB access to the test data? If so, the authors should change that (or collect new test data?) to ensure a fair evaluation.

---

### Official Review · AnonReviewer1 · 2020-10-29
**Not persuasive enough in translating short expressions**

**Rating:** 4
**Confidence:** 4

**Review:**

This paper proposes another variant of phrase-based MT for African languages,
involving native speakers for manual annotations.
Instead of just using subwords or statistical phrase identification, the
authors propose to use the intuition of native speakers for translating African
Fon languages into French (and vice versa).
According to their experiments, BLEU and other indexes significantly improved
over standard IBM-1 phrase-based machine translation.

However, from the description and examples in this paper, I have a little doubt
for this improvement:

- For creating the aligned corpus, the authors say that they chose only short
expressions, namely 1-6 words. According to the results shown in Table 1,
this essentially amounts to simply memorizing frequent idiomatic phrases.
Therefore, improvements with this kind of human intervention over such an
easy sentences is basically trivial. Of course, the paper says that the test
data comprises of long and complex sentences; but the examples are not, thus
I cannot identify whether the problem is really difficult or not.

- Even if the proposed human annotation is effective, that does not seem to
leverage characteristic property of African languages. In Section 3, "un d'o
ganji" has an ambiguities about "un", but this kind of ambiguity of a word is
shared by almost all the other languages (imagine translating "given" in a
conditional proposition). The property of African Fon languages, such as
diacritics and affixation, are not used here.

Finally, the proposed annotation algorithm in page 3 seems to quite vague to
me. Where v came from? If w is a word, what is the meaning of "w \subseteq v"?
Also, this algorithm seems to use a simple longest match: however, in many
cases the usage of a word is only clear using succeeding words; i.e. some
forward-backward algorithm is necessary for correct identification of a phrase.

That being said, I strongly agree with the authors that neural machine
translation of African low-resourced language is important. I hope that the
authors would add more persuasive results and analysis to realize a practical
translation of Fon languages.

---

### Decision · Program_Chairs · 2021-01-07
**Final Decision**

**Decision:**

Reject

**Comment:**

The authors investigate different tokenization methods for the translation between French and Fon (an African low-resource language). Low-resource machine translation is a very important topic and it is great to see work on African languages - we need more of this!

Unfortunately, the reviewers unanimously agree that this work might be better suited for a different conference, for example LREC, since the machine learning contributions are small. The AC encourages the authors to consider submitting this work to LREC or a similar conference.